# Analysis, Design, and Experimental Research of a Novel Wheelchair-Stretcher Assistive Robot

**Lingfeng Sang** [1,2,3,*] **, Masayuki Yamamura** [2] **, Fangyan Dong** [3] **, Zhongxue Gan** [3] **, Jianzhong Fu** [4] **, Hongbo Wang** [5] **and Yu Tian** [5]

1   College of Mechanical and Electrical Engineering, Ningbo Polytechnic, Ningbo 315800, China
2   School of Computing, Tokyo Institute of Technology, Kanagawa 226-8502, Japan; my@c.titech.ac.jp
3   Ningbo Intelligent Manufacturing Industry Research Institute, Yuyao 315400, China;
    dongfangyan@iimi.org.cn (F.D.); ganzhongxue@fudan.edu.cn (Z.G.)
4   College of Mechanical Engineering, Zhejiang University, Hangzhou 310027, China; fjz@zju.edu.cn
5   Parallel Robot and Mechatronic System Laboratory of Hebei Province, Yanshan University,
    Qinhuangdao 066004, China; hongbo_w@ysu.edu.cn (H.W.); ysusirtian@163.com (Y.T.)
*   Correspondence: sanglingfeng@163.com



**Featured Application: This novel robot could be applied in hospital, family and nursing home to help the elderly complete the basic daily activities and move patient autonomously.**

**Abstract:** Life care for disabled or semi-disabled elderly has become an increasingly common problem in society. In this paper, a novel wheelchair-stretcher assistive robot, which can meet the physiological needs of patients, is investigated and designed. The following tasks are conducted: (1) the mecanum wheel is adopted as the executive device of the walking mechanism, and its kinematics is analyzed in detail. (2) A five-link mechanism with single degree of freedom is proposed to realize the folding motion of the robot. Through the minimum conclusive area method, the optimal sizes of the armrests link and the side link are 507.9, and 332.5 mm, respectively. Based on the force analysis of the linkage mechanism, six torsion springs and RV (rotate vector) reduction motor are used as the driving device, which reduces the driving torque of the motor. (3) Based on the STM32 (STMicroelectronics 32-bits Microcontroller) chip, and combined with the theoretical analysis, the mechanical structure and the control system of the whole prototype are designed, and the feasibility of each module is verified by experimental research. The results confirm that the proposed robot has good performance, and that the control algorithm for the walking mechanism and the lifting mechanism are suitable.

**Keywords:** wheelchair-stretcher assistive robot; mechanical structure; kinematics analysis; parameter optimization; control system

## 1. Introduction

It is generally known that population aging is a global phenomenon because of the increasing levels of health care. In 2019, the number of people aged 65 years or over has become 703 million in the world, and it is estimated to double in 2050 [1,2]. With the aging population, there are more and more disabled and semi-disabled elderly people. Despite that bedridden elderly needs to go to the toilet in a wheelchair or go out for some fresh air, it is impossible to do on their own [3].

Researchers have done a lot of work on enabling the elderly to smoothly move between bed and wheelchair. For instance, bed chair assist robot "Resyone" is developed by Panasonic Company in Japan [4]; a bed-chair robot system "AgileLife", that can be docked with wheelchair, is made by NextHealth Company [5]; a multi-functional wheelchair of assisting the elderly to get up and down

the bed is proposed by Harbin Engineering University [6]; an integrated multi-functional nursing bed for the bedridden elderly with weak body but normal intelligence is developed by Institute of Robotics of Beijing University of Aeronautics and Astronautics and Xinsong Robot Automation Co., Ltd. [7,8]. By changing the structure of the bed, the above studies enable patients to move between the bed and the wheelchair. There is also some research on the transport device from bed to stretcher, such as a patient transport device developed by SA.VIR Company in Italy [9], taxiing air cushion designed by Getinge Company in Sweden [10], "PowerNurse" transfer equipment invented by Astir Technologies Company in American [11], SE series medical electric transfer vehicle developed by Ningbo Kai Medical Technology Co., Ltd. [12], a transfer robot researched by the Institute of Health Equipment of the Academy of Military Medical Sciences of the Chinese people's Liberation Army [13], and intelligent transport robot for patients designed by Yanshan University [14].

In addition, some researchers use humanoid robots to study the transfer and transport of patients, such as lifting transfer robot "RIBA", invented by the Institute of Physical and Chemical Research in Japan [15], the wounded transfer robot BEAR developed by Vecna Company in the United States [16], and the humanoid back-hugging transfer robot "Baize" designed by Hebei University of Technology [17].

In summary, we can see that:

(1) The technology of transport devices in moving patients from bed to stretcher, or from stretcher to bed, is more mature; but it is difficult to transform the stretcher to a wheelchair.

(2) The research on the multi-function nursing bed mainly focuses on the change of bed structure. There are two main forms; one is that the middle part of the bed structure is changed directly into a wheelchair; the other is that one side of the bed structure is transformed into a wheelchair. However, for these two methods, not only the sheets need to be customized, but also the mattress under the body needs to be moved with the patient which resulted in a deterioration of the patient's experience.

(3) The research technology of humanoid assistive robot is advanced, but it is expensive and is far away from industrial application.

(4) It has always been a difficult problem for patients to go to the toilet. Although the research institutions have put forward some ideas, they have not been able to successfully develop the right products.

Based on these, a low-cost and practical wheelchair-stretcher assistive robot is proposed. The robot could transform between wheelchair and stretcher. In wheelchair form, it could move autonomously, so that the elderly can easily complete the basic daily activities such as going to the bathroom and outdoors. In stretcher form, it could automatically adjust the height and dock with the bed. The robot can also work with transport device to move patient between bed and stretcher which promote comfort and prevent harm to patients.

## 2. Mechanical Structure Design of the Wheelchair-Stretcher Assistive Robot

The product is determined on the basis of user requirements and corresponding design standards. As wheelchair and stretcher belong to medical equipment, the designed robot should meet the corresponding medical device standards [18,19]. In order to ensure that the human-machine-environment system can be in line with human movement habits, physiological habits and psychological habits, and to ensure that people can use robots to carry out all kinds of movements safely, comfortably and efficiently. Therefore, the theory of ergonomics is adapted as the other design standard [20,21]. Based on these two standards, the proposed robot is designed.

The robot consists of two main parts, namely, the design of the mechanical structure and the design of the control system. The design of the mechanical structure includes the design of the folding mechanism, the walking mechanism, the lifting mechanism and the toilet mechanism; the control

system design includes the main control unit, drive unit, and remote control unit, as shown in Figure 1. The prototype is shown in Figure 2.

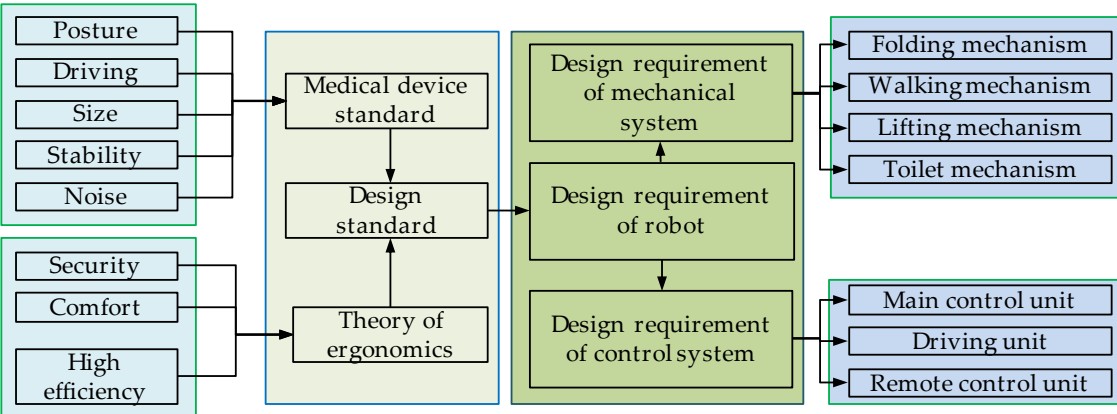

**Figure 1.** Designing scheme of the wheelchair-stretcher assistive robot.

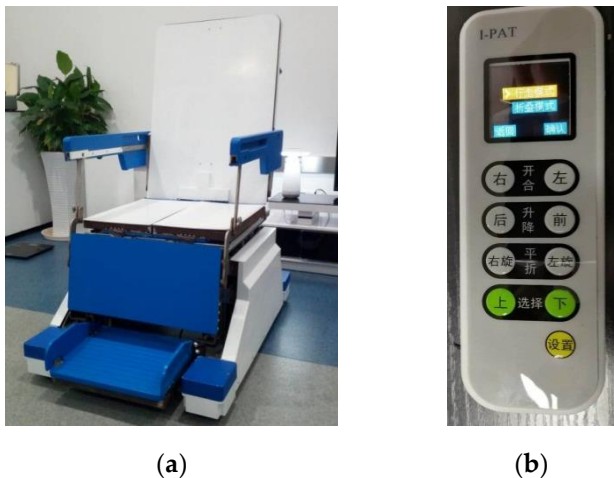

(**a**)                    (**b**)

**Figure 2.** The wheelchair-stretcher assistive robot; (**a**) prototype of the robot; (**b**) remote control of the robot.

*2.1. Kinematics Analysis and Structure Design of the Walking Mechanism*

Kinematics play a crucial role in the selection of mechanism design parameters. First, the kinematics of walking mechanism is analyzed in detail.

2.1.1. Kinematics Analysis of the Walking Mechanism

The mecanum wheel, a kind of omnidirectional wheel, is selected for the design of the walking mechanism. The reason is that a combination of three or more mecanum wheels can move in any direction on a flat surface [22]. The typical four-wheel longitudinal symmetrical layout structure is applied in the walking mechanism, as shown in Figure 3. The global coordinate frame $\{o_{xy}\}$ is assigned on the center of the walking platform and the velocity of the center point is described as $\begin{bmatrix} v_x & v_y & w_z \end{bmatrix}^{\mathrm{T}}$.

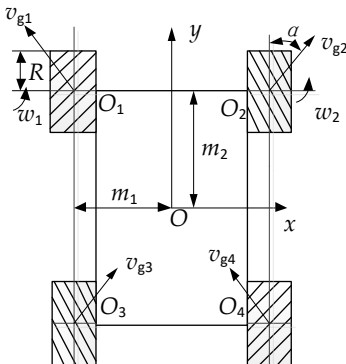

**Figure 3.** Layout analysis of mecanum wheel.

Assuming that the roller does not skid against the ground and the platform moves on the horizontal plane, the velocity $v_4$ of the fourth wheel can be established by kinematics analysis:

$$v_4 = \begin{bmatrix} 0 & -\sin\alpha \\ R & \cos\alpha \end{bmatrix} \begin{bmatrix} w_4 \\ v_{g4} \end{bmatrix}, \tag{1}$$

where $R$ is the radius of the Mecanum wheel; $\alpha$ is the bias angle of the roller and the diagonal line in Figure 3 indicates the layout of the rollers in contact with the ground; $w_4$ is the angular velocity of the fourth wheel; $v_{g4}$ is velocity to perpendicular to the roller.

Meanwhile, in the global coordinate system, the following equation can be obtained,

$$v_4 = \begin{bmatrix} 1 & 0 & m_2 \\ 0 & 1 & m_1 \end{bmatrix} \begin{bmatrix} v_x \\ v_y \\ w_z \end{bmatrix} \tag{2}$$

where $m_1$, $m_2$ are horizontal and vertical distance from the center of the platform to the center of the wheel respectively.

Combining Equations (1) and (2), we can obtain:

$$\begin{bmatrix} 0 & -\sin\alpha \\ R & \cos\alpha \end{bmatrix} \begin{bmatrix} w_4 \\ v_{g4} \end{bmatrix} = \begin{bmatrix} 1 & 0 & m_2 \\ 0 & 1 & m_1 \end{bmatrix} \begin{bmatrix} v_x \\ v_y \\ w_z \end{bmatrix}. \tag{3}$$

Similarly, the kinematics expression from the first wheel to the three wheels can be computed; finally, the angular velocity vector $w$ for the four wheels can be described as,

$$w = \begin{bmatrix} w_1 \\ w_2 \\ w_3 \\ w_4 \end{bmatrix} = J_1 \begin{bmatrix} v_x \\ v_y \\ w_z \end{bmatrix}, \tag{4}$$

where

$$J_1 = \begin{bmatrix} \frac{1}{R\tan\alpha} & \frac{1}{R} & \frac{-(m_1\tan\alpha+m_2)}{R\tan\alpha} \\ \frac{-1}{R\tan\alpha} & \frac{1}{R} & \frac{m_1\tan\alpha+m_2}{R\tan\alpha} \\ \frac{-1}{R\tan\alpha} & \frac{1}{R} & \frac{-(m_1\tan\alpha+m_2)}{R\tan\alpha} \\ \frac{1}{R\tan\alpha} & \frac{1}{R} & \frac{m_1\tan\alpha+m_2}{R\tan\alpha} \end{bmatrix}.$$

The angular velocity is obtained by the transmission of the motor and the gearing; therefore, the corresponding relationship between the rotational velocity of the motor and the velocity of the platform can be obtained as follows,

$$n = \frac{w}{2\pi m},$$ (5)

where *m* is the reduction ratio of the mecanum wheels.

### 2.1.2. Structure Design of the Walking Mechanism

Based on the computing of the walking mechanism and the layout arrangement of the mecanum wheel, the mechanical structure of the walking mechanism, which is installed on the support plate of the seat, is designed in detail and shown in Figures 4–6. The left part of the walking mechanism is similar to the right part. The mecanum wheel, which is mounted in the case of the walking wheel, is connected to NMRV040 turbine worm reducer by the synchronous wheel and the synchronous belt; and the motor is linked to the turbine worm reducer by P52HA planetary reducer. Therefore, the mecanum wheel could easily get power from the motor.

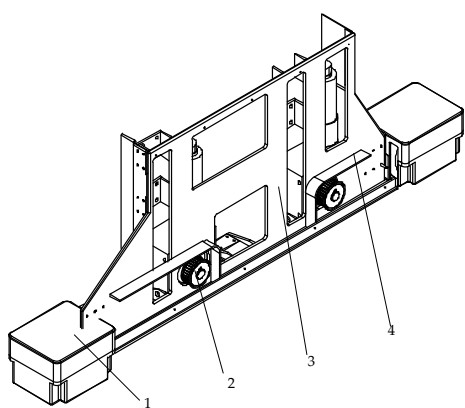

**Figure 4.** The first design drawing of the right walking mechanism; 1—case cover of the walking wheel, 2—synchronous belt pulley, 3—seat support plate, 4—belt guard.

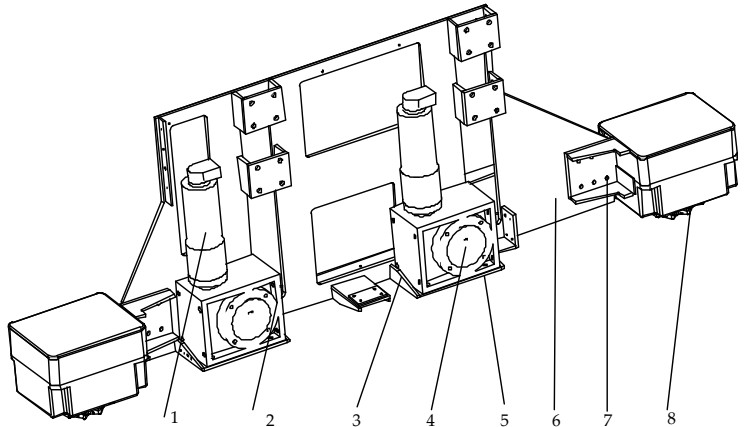

**Figure 5.** The second design drawing of the right walking mechanism; 1—36SYK71 motor and P52HA planetary reducer, 2—NMRV040 turbine worm reducer, 3—rib plate, 4—shaft, 5—base plate of the motor, 6—lower support plate of the seat, 7—connecting plate of the walking wheel, 8—mecanum wheel.

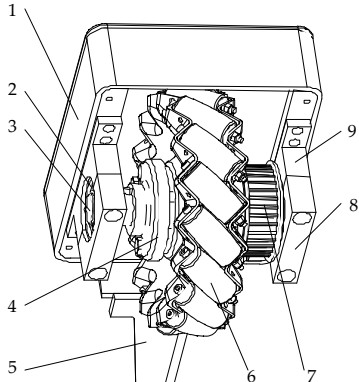

**Figure 6.** The mechanical structure design of the Mecanum wheel; 1—case cover of the walking wheel, 2—bear, 3—shaft of the Mecanum wheel, 4—connectting plate of the Mecanum wheel, 5—connecting plate of the walking wheel, 6—the Mecanum wheel, 7—HTD5 driven belt pulley, 8—the upper cover of the bearing block, 9—the lower cover of the bearing block.

### 2.2. Kinematics, Dynamics Analysis and Optimization of the Folding Mehanism

#### 2.2.1. Kinematics Analysis and Parameters Optimization of the Folding Mechanism

Folding mechanism is the key to realizing the safe and smooth switching between wheelchair and stretcher. Analyzing the movement characteristic of the wheelchair-stretcher assistive robot and researching the references [23–27], the corresponding mechanism is proposed, as shown in Figure 7.

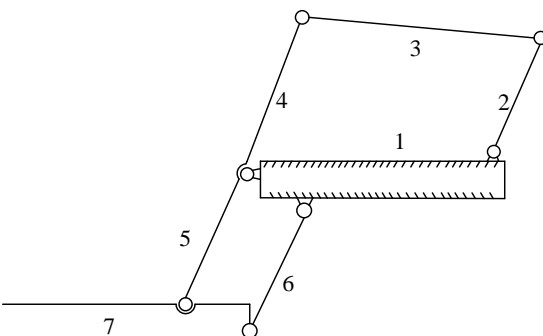

**Figure 7.** Schematic diagram of folding mechanism; 1—the seat plate, 2—side link 1, 3—armrests, 4—side link 2, 5—calf support plate, 6—connecting rod, 7—foot pedal.

There is one rack, five motion components and seven revolute joints in the mechanism. On the basis of the computing formula of the degree of freedom for the planar linkage mechanism [28], the following equation can be obtained:

$$dof = 3n - 2P_L - P_H = 1 \tag{6}$$

where $n$ is the number of motion components, $P_L$ is the number of lower pair, and $P_H$ is the number of higher pair.

Parallelogram structure composed by the calf support plate, foot pedal, connecting rod and the seat plate has no effect on the motion of the whole folding mechanism. Therefore, the motion of the quadrilateral structure, which consisted of the seat plate, side link 1, armrests and side link 2, is only analyzed.

In order to determine the structural size of the mechanism, the minimum conclusive area, which is the smallest plane region produced in a period for the motion of the folding mechanism, is used as the optimization function [29]. The movement diagram of the whole mechanism is shown in Figure 8.

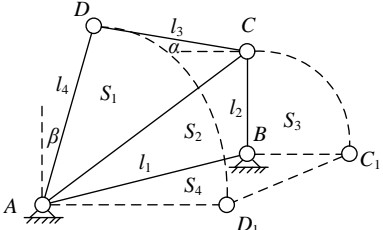

**Figure 8.** Schematic diagram of the planar four-bar linkage.

The two positions shown in Figure 8 are the limit positions of the movement of the planar linkage, and the region formed by the two limit positions is the smallest region of the planar linkage and can be described as:

$$S = S_1 + S_2 + S_3 + S_4. \tag{7}$$

In this formula, area $S_1$ of $\triangle ADC$ is given as,

$$S_1 = \sqrt{p(p - l_4) \cdot (p - l_3) \cdot (p - l_2)} \tag{8}$$

where $l_2 = 290$ mm is the length of side link 1. The optimal parameter $l_3$ is the length of the armrests. The optimal parameter $l_4$ is the length of side link 2. And $p = \frac{l_3 + l_4 + AC}{2}$.

Area $S_2$ of $\triangle ABC$ is expressed as,

$$S_2 = \frac{1}{2} l_1 \cdot l_2 \cdot \sin \angle ABC \tag{9}$$

where $l_1 = 555$ mm is the length of the seat plate.

Area $S_3$ of sector $CBC_1$ is written as:

$$S_3 = \frac{1}{4} \pi l_2^2. \tag{10}$$

Area $S_4$ of trapezoid $ABC_1D_1$ is described as:

$$S_4 = \frac{(l_1 + l_2) \cdot |AB|_y}{2}. \tag{11}$$

In the meantime, the relationship expressions between $l_3, l_4$ and $\alpha, \beta$ can be determined as follows,

$$
\begin{aligned}
l_4 \cdot \sin \beta + l_3 \cos \alpha &= l_1 \cos(\angle D_1 AB), \\
l_4 \cdot \cos \beta - l_3 \sin \alpha &= l_1 \sin(\angle D_1 AB) + l_2
\end{aligned} \tag{12}
$$

where $\alpha$ is the angle between the armrests link and the horizontal direction. $\beta$ is the angle between the side link 2 and the vertical direction. $\angle D_1 AB$ is the angle between the side link 2 in the limit position and the seat plate.

Simplifying Equation (12), the following equations can be derived:

$$
\begin{aligned}
l_3 &= \frac{l_1 \cos(\angle D_1 AB) \cos \beta - l_1 \sin(\angle D_1 AB) \sin \beta - l_2 \sin \beta}{(\sin \alpha \sin \beta + \cos \alpha \cos \beta)}, \\
l_4 &= \frac{l_1 \sin \alpha \cos(\angle D_1 AB) + l_1 \cos \alpha \sin(\angle D_1 AB) + l_2 \cos \alpha}{\sin \alpha \sin \beta + \cos \alpha \cos \beta}.
\end{aligned} \tag{13}
$$

Combining the Equations (8)–(13) into Equation (7), the expression for *S* can be obtained. Meanwhile, the condition that the mechanism has a crank can be determined as:

$$
\begin{cases}
l_1 + l_2 \leq l_3 + l_4 \\
\quad l_2 \leq l_3 \\
\quad l_2 \leq l_4
\end{cases}
. \tag{14}
$$

The angles $\alpha$, $\beta$ are set in the range $0 \leq \alpha \leq 10°$ and $0 \leq \beta \leq 10°$, respectively. Based on the Equation (7) and relationships (14), the change in *S* can be obtained when the parameters $l_3$ and $l_4$ change, as shown in Figure 9. Observing this figure, it can be found that the area *S* is the smallest and its value is $2.42 \times 10^5$ mm$^2$ when the value of the parameters $l_3$, $l_4$ are 507.9 mm, and 332.5 mm, respectively.

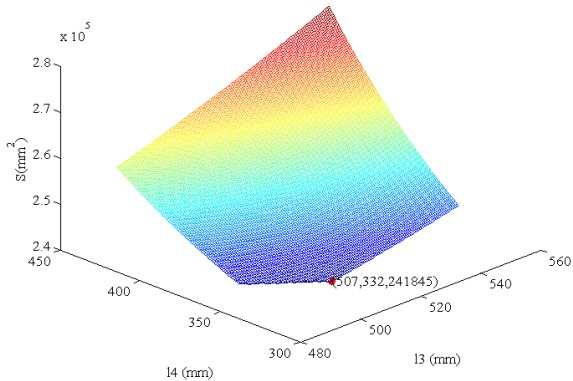

**Figure 9.** Parameter optimization of the folding mechanism.

### 2.2.2. Force Analysis of the Folding Mechanism

The force research for the folding mechanism can provide a basis for the selection of the motor, as well as the setting, and optimization of the control parameters. In this section, force analysis is conducted in detail. The stress analysis diagram is shown in Figure 10.

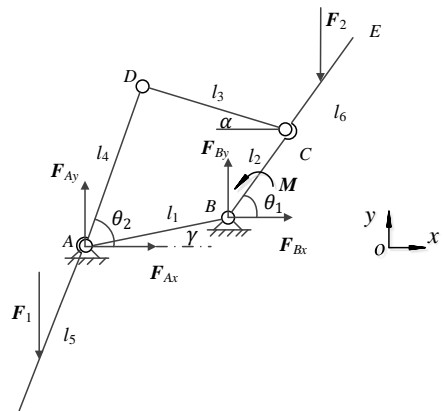

**Figure 10.** Force analysis of the folding mechanism.

According to the stress analysis diagram, the force equation in *x*-axis direction can be established as,

$$
F_{Ax} + F_{Bx} = 0 \tag{15}
$$

where $F_{Ax}$ is the force of the horizontal direction on fulcrum *A*, and $F_{Bx}$ is the force of the horizontal direction on fulcrum *B*.

In the meanwhile, the force equation in $y$-axis direction also can be expressed as,

$$F_{Ay} + F_{By} = F_1 + F_2 \tag{16}$$

where $F_{Ay}$ is the force of the vertical direction on fulcrum $A$, $F_{By}$ is the force of the vertical direction on fulcrum $B$, $F_1$ is the force on the calf support plate, and $F_2$ is the force on the back plate.

So resultant moment on fulcrum $A$ can be obtained as,

$$F_1 \cdot \frac{1}{2} \cdot l_5 \cdot \sin \theta_2 + F_{By} \cdot l_1 \cdot \cos(e) - F_2 \cdot \left[ \left( l_2 + \frac{1}{2} l_6 \right) \cdot \cos \theta_1 + l_1 \cdot \cos(e) \right] + M = 0 \tag{17}$$

where $l_5$ is the length of the calf support plate, $l_6$ is the length of the back plate, $\theta_1$ is the angle between the back plate and direction of $x$-axis, $\theta_2$ is the angle between the side link 2 and direction of $x$-axis, $M$ is the torque of the motor, $\gamma$ is the angle between the seat plate and the direction of $x$ axis.

Correspondingly, the resultant moment on the fulcrum $B$ can be described as:

$$\begin{aligned} F_1 \cdot \left( \tfrac{1}{2} \cdot l_5 \cdot \cos \theta_2 + l_1 \cdot \cos(e) \right) - F_{Ay} \cdot l_1 \cdot \cos(e) \\ - F_2 \cdot \left( l_2 + \tfrac{1}{2} l_6 \right) \cdot \sin \theta_1 + F_{Ax} \cdot l_1 \cdot \sin(e) + M = 0 \end{aligned} \tag{18}$$

In the folding mechanism process, the force applied at the center of the calf support plate can be equaled by the weight of human lower limbs and mechanism weight of this part; and the force applied at the center of the back plate can be equaled by the weight of the upper body and the mechanism weight in this part. According to the standards of ergonomics and adult body size in China [30,31], the values of parameters $l_5$, $l_6$, $F_1$, $F_2$ are set as 490 mm, 510 mm, 300 N, and 600 N, respectively.

Through the force analysis of $l_4$ and $l_5$, the resultant moment on point $D$ can be obtained as:

$$F_1 \cdot \left( \frac{1}{2} l_5 + l_4 \right) \cdot \cos \theta_2 + F_{Ax} \cdot l_4 \cdot \sin \theta_2 - F_{Ay} \cdot l_4 \cdot \cos \theta_2 = 0. \tag{19}$$

Through the force analysis of $l_2$ and $l_6$, the resultant moment on point $C$ can be obtained as:

$$- F_2 \cdot \left( \frac{1}{2} \cdot l_6 + l_2 \right) \cdot \cos \theta_1 + M - F_{By} \cdot l_2 \cdot \cos \theta_1 + F_{Bx} \cdot l_2 \cdot \sin \theta_1 = 0. \tag{20}$$

The following geometric relationships between $\theta_1$ and $\theta_2$ can be described as:

$$\begin{cases} l_4 \cdot \cos(\theta_2) + l_3 \cdot \cos \alpha - l_2 \cdot \cos \theta_1 = l_1 \cdot \cos(e), \\ l_1 \cdot \sin(e) = l_4 \cdot \sin(\theta_2) - l_3 \cdot \sin \alpha - l_2 \cdot \sin \theta_1. \end{cases} \tag{21}$$

Merging Equations (15)–(21) by using Matlab software to program, the torque $M$ required by the folding mechanism can be solved as:

$$M = 162 \cos \theta_1. \tag{22}$$

It is found that the maximum torque is 162 N·m. If such a high output torque is chosen, the size and power of the motor are too big for the folding mechanism. It is necessary to add a booster device to the mechanical structure.

### 2.2.3. Driving Device Computation of the Folding Mechanism

According to the above analysis, the degree of freedom for the folding mechanism is 1, so only one driving motor is needed to complete the mechanical motion. It is proposed that 24 V DC gear motor is adopted as the main driving power, and the torsion spring group as the auxiliary driving power for the motion of the folding mechanism. Since the maximum torque of the gear motor is about 30 N·m, the torsion spring group needs to overcome all the remaining torques; therefore, the designed

torsional spring group includes 6 torsional springs, each of which can withstand a maximum torque of 22 N·m and a maximum torsional angle $\varphi_m = 90°$.

According to the character of the load, the spring material is determined to be 60Si2MnA, elastic modulus $E$ is 206 GPa, ultimate tensile strength $\sigma_b$ is 1667 MPa, allowable bending stress $[\sigma_b]$ is 1334 MPa.

Based on these parameters, the required spring steel diameter $d$ can be calculated as,

$$d = \sqrt[3]{\frac{32T_nK_1}{\pi[\sigma_b]}} \approx 5.7\text{mm} \tag{23}$$

where $K_1 = 1.15$ is coefficient of curvature, $T_n = 22$ N·m is maximum working torque. According to the design requirements, the value of the steel wire diameter $d$ should be rounded, so 6 mm should be taken.

The effective coil number $n$ of spring can be obtained by the following equation:

$$n = \frac{Ed^4\varphi_m}{3667D(T_n - T_1)} \approx 7.4 \tag{24}$$

where $T_1 = 0$, $D$ is the mean diameter of coil and $D = 40$ mm. According to the actual working situation, the effective winding number of spring is selected as 7.

The stiff of the spring is expressed as:

$$T' = \frac{Ed^4}{3667D \cdot n} = 0.26\text{N·m}. \tag{25}$$

Similarly, the free length of the spring can be computed as

$$H_0 = n \cdot t + d = 51.5\text{mm} \tag{26}$$

where $t = 6.5$ mm.

According to the force analysis of the folding mechanism and the design analysis of the torsional spring, the torque required by the motor can be calculated as,

$$T_m = M - T \tag{27}$$

where $T_m$ is the torque of the motor, $T$ is the torque of the spring and $T = T'\varphi$, $\theta_1 = 90 - \varphi$.

According to the Equations (22), (25), and (27), the torque variation curve can be plotted, as shown in Figure 11. It is found that the torque of the motor is maximum in $\varphi = 57°$ place and its value is $T_m = 46$ N·m. According to this parameter, the performance parameters of the worm gear reducer motor can be determined in shown in Table 1.

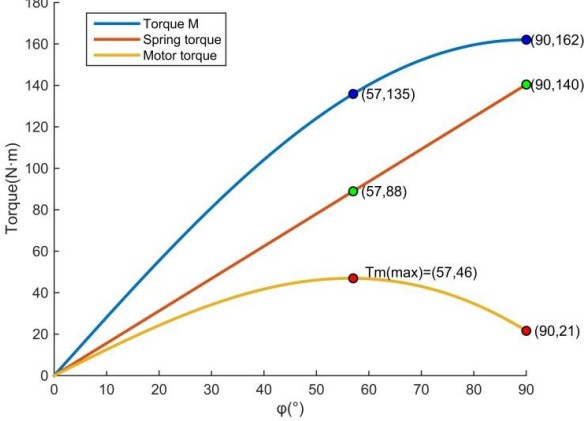

**Figure 11.** Torque change of folding mechanism.

**Table 1.** Parameter list of RV (rotate vector) reduction motor.

| Voltage/V | Reducer | Rated Torque/(N·m) | Rated Power/W | Rated Revolution/(r·min$^{-1}$) |
|---|---|---|---|---|
| 24 | 1/862 | 48 | 200 W | 2 |

### 2.2.4. Design of the Folding Mechanism

Through the above analysis and computing, the designed folding mechanism is composed of the foot pedal, the calf support plate, the seat plate and the back plate, as shown in Figure 12. These four sections are assembled together by the bolt and the hinge. All the components of the folding mechanism are shown in Figure 13, which is axonometric drawing.

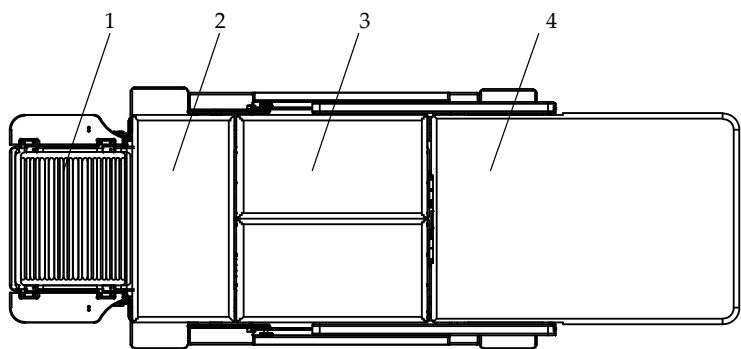

**Figure 12.** Vertical view of folding mechanism; 1—the foot pedal, 2—calf support plate, 3—the seat plate, 4—the back plate.

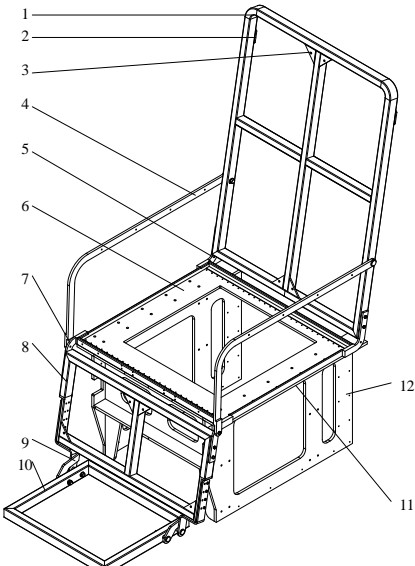

**Figure 13.** Axonometric view of folding mechanism; 1—the pipe structure of the back plate, 2—support of the back plate, 3—connecting plate, 4—armrests, 5—backrest fitting, 6—cushion support, 7—leg fitting, 8—the pipe structure of the leg, 9—pedal connecting strip, 10—the pipe structure of the pedal, 11—the upper support plate of the seat, 12—the support plate in the seat side.

Figure 14 is the back diagram of the folding mechanism. In Figure 14, the installation method and position of the six springs are pointed out, and the installation method of the worm gear and worm reducer motor and torsion spring is given.

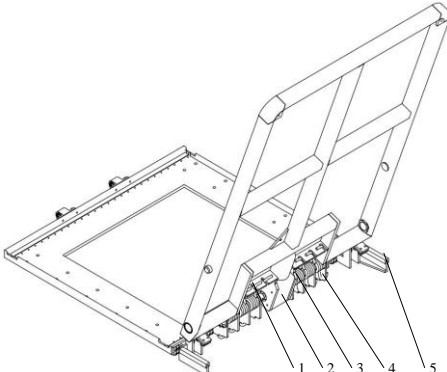

**Figure 14.** Back mechanism diagram of the wheelchair-stretcher robot; 1—the support plate of the spring, 2—motor, 3—spring seat, 4—torsional spring, 5—Rubber cushion for the backrest.

*2.3. Design of the Lifting Mehanism and Toilet Mechanism*

When docking with the bed or the toilet, the height needs to be often adjusted; so the lifting mechanism is designed. Considering the docking demand in the family environment, the height of the lifting mechanism is the range 480 mm–680 mm. For safety, 24 V DC power is supplied. In the lifting process, considering the compact structure and the balance, linear actuators are arranged on both sides for pushing and two linear guideways on each side are equipped to realize the vertical lifting and ensure good parallelism between the two linear actuators. In order to realize the synchronous control of both sides, linear displacement sensor is used to control the real-time position. The design diagram of the whole mechanism is shown in Figures 15 and 16.

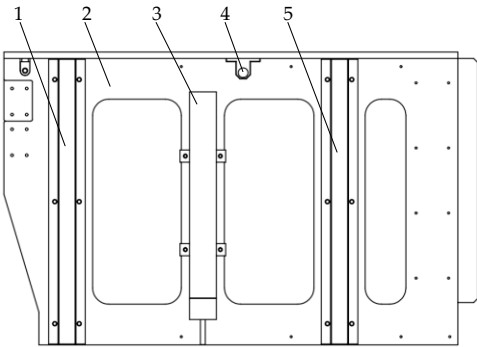

**Figure 15.** The upper support plate of the seat; 1—linear guide on the left, 2—the upper support plate of the seat, 3—Linear displacement sensor, 4—the seat of the linear actuator, 5—linear guide on the left.

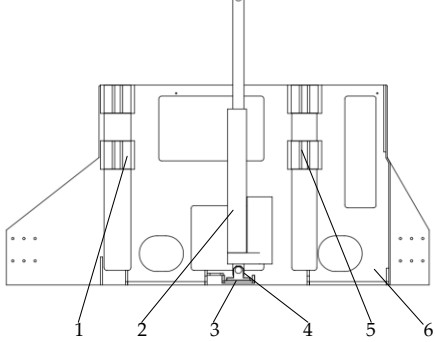

**Figure 16.** The lower support plate of the seat; 1—guide block on the left, 2—the linear actuator, 3—welding support of the linear actuator, 4—the seat of the linear actuator, 5—guide block on the right, 6—the lower support plate of the seat.

In order to achieve the patients or the elderly toilet, the push and pull mechanism is used. Firstly, the height of the seat plate is adjusted, and then the left pad and the right pad in the toilet mechanism are pulled to the sides to go to the toilet for the elderly. When the elderly finish the toilet, the left pad and the right pad are pushed to the original position. The movement diagram and design drawing for the toile mechanism are shown in Figures 17 and 18.

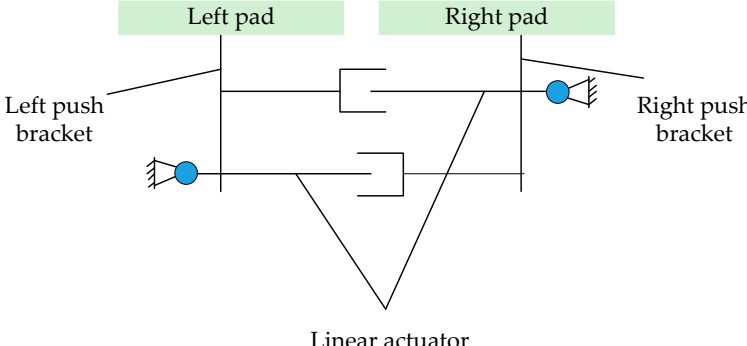

**Figure 17.** The movement diagram of the toilet mechanism.

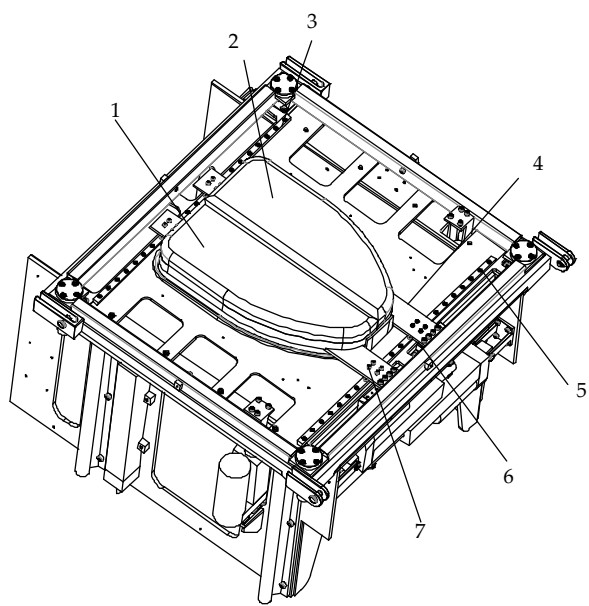

**Figure 18.** The design drawing of the toilet mechanism; 1—the left pad, 2—the right pad, 3—back slide, 4—the right plate, 5—front slide, 6—the front sliding block, 7—the left plate.

## 3. Control System Design of the Wheelchair-Stretcher Assistive Robot

The control system of the wheelchair-stretcher assistive robot includes 4 motors for the walking mechanism, 2 linear actuators for toilet mechanism, 2 linear actuators for lifting mechanism, and 1 motor for folding mechanism and several sensors. The whole system is controlled by a remote control. The remote controller transmits instructions to the main control board, and the radio frequency board, connected to the main control board, receives signals and performs corresponding actions according to the control instructions. The hardware control block diagram of the whole system is shown in Figure 19.

Toilet mechanism control and folding mechanism control are relatively simple, both are open-loop position control, which is not described here. The control of the walking mechanism and lifting mechanism involves the position feedback problem, and the control process is relatively complex. Therefore, a detailed analysis is made on these two parts.

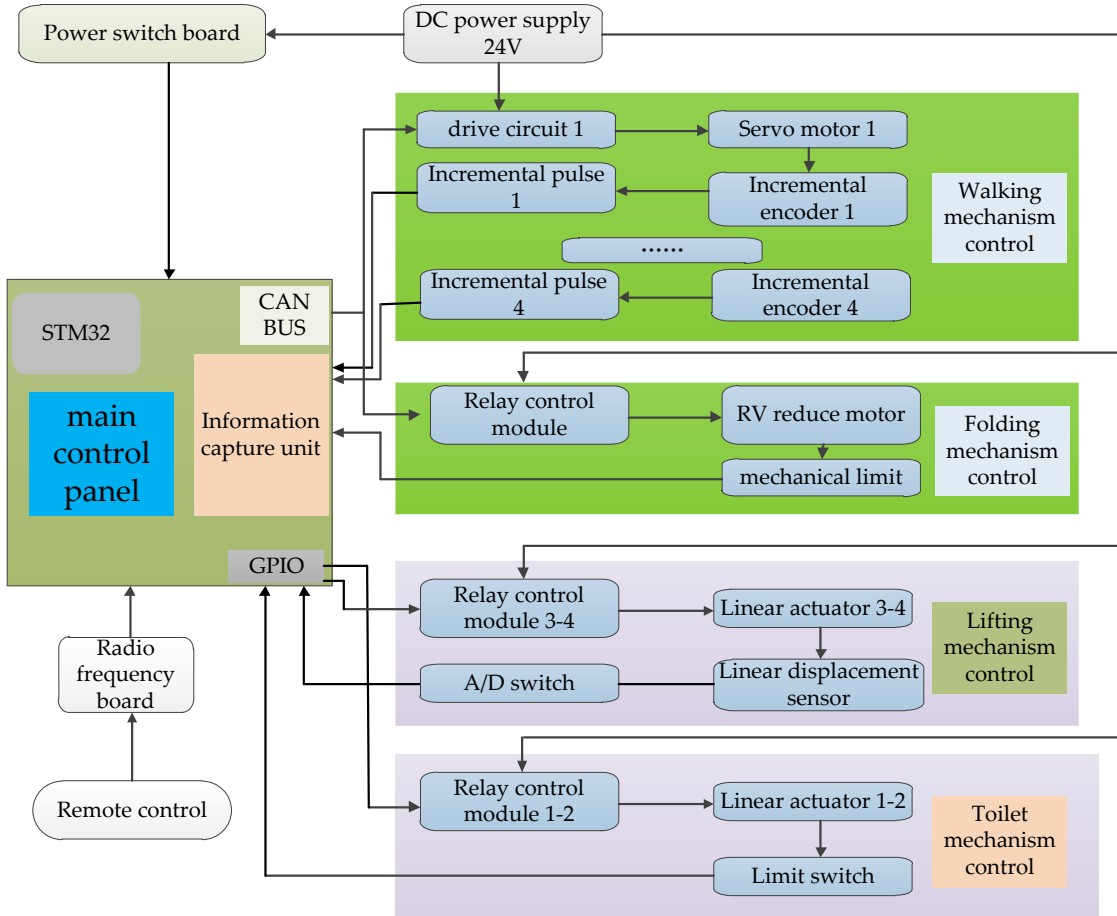

**Figure 19.** Control block diagram of the wheelchair-stretcher assistive robot. STM32: STMicroelectronics 32-bits Microcontroller; GPIO: general purpose input/output; RV: rotate vector.

## 3.1. Control of the Walking Mechanism

The walking mechanism mainly realizes six kinds of motions, which are forward, backward, left, right, turning both sides. The main control board transmits four PWM pulses respectively to control the walking mechanism. In order to ensure the accuracy of the moving process, the compound position control method is adopted to control six kinds of motion. The so-called composite motion adopts velocity control between two discrete position nodes and detects the position at the same time. The position error of this node is taken as part of the theoretical velocity control of the next node and the theoretical velocity is corrected. The control block diagram is shown in Figure 20.

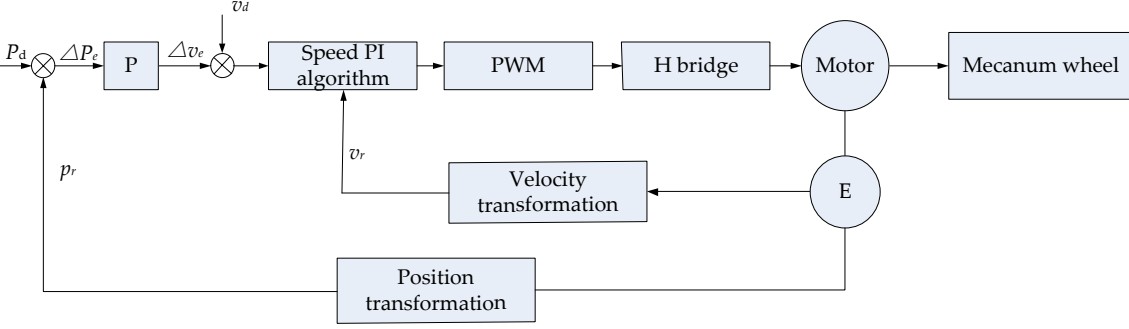

**Figure 20.** Control block diagram of compound position control method.

According to the figure, the position error of this node due to speed control is,

$$\Delta p_e = p_d - p_r \tag{28}$$

where $p_d$ is the reference value of the position, $p_r$ is the real value of position detected by the controller.

The velocity of the next node is:

$$v'_d = P\Delta p_e + v_d. \tag{29}$$

### 3.2. Control of the Lifting Mechanism

The control block diagram of the lifting mechanism is shown in Figure 21. This part adopts the position closed-loop control, and the modified *PWM* value can be obtained through the transformation of real-time monitoring position information.

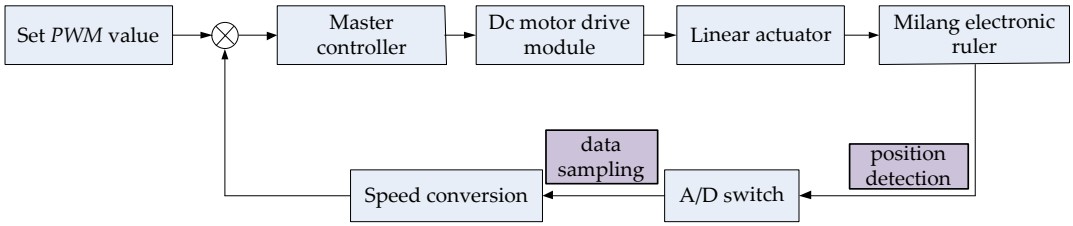

**Figure 21.** Control block diagram of the lifting mechanism.

The process is as follows: firstly, we assume that the same *PWM* values of the two linear actuators are given as,

$$PWM_1 = PWM_2 \tag{30}$$

where $PWM_1$ is the value of the linear actuator $L_1$, $PWM_2$ is the value of the linear actuator, $L_2$.

At time $t$, the output voltage values of the two electronic rulers are $U_1$, $U_2$. Since the output voltage of the linear displacement sensor is linearly related to the moving displacement, the moving displacement of the two electronic rulers can be described as,

$$\begin{aligned} S_1 &= U_1 \cdot K_d, \\ S_2 &= U_2 \cdot K_d \end{aligned} \tag{31}$$

where $K_d$ is the proportional coefficient.

The velocity difference of the two linear actuators in time $t$ can be computed as,

$$\Delta V = V_1 - V_2 = \frac{\Delta S}{t} = \frac{S_1 - S_2}{t}. \tag{32}$$

where $V_1$ is the velocity of the linear actuator $L_1$, $V_2$ is the velocity of the linear actuator $L_2$.

The data of the linear actuator $L_1$ is used as the reference value, and the velocity of the linear actuator $L_2$ can be modified as:

$$V_2 = \Delta V + V_1. \tag{33}$$

Since the velocity of the motor is directly proportional to the duty cycle, it can be described as,

$$V = K_\gamma \cdot \gamma \tag{34}$$

where $K_\gamma$ is the assuming proportional coefficient, $\gamma$ is the duty cycle.

According to Equations (33) and (34), the duty cycle of the linear actuator $L_2$ can be modified as:

$$\gamma_2 = \frac{V_2}{K_\gamma} = \frac{\Delta V}{K_\gamma} + \frac{V_1}{K_\gamma}. \tag{35}$$

The position closed-loop control for the lifting mechanism is conducted by the STM32 chip. So PWM pulse modulation mode and ADC mode of STM32 are used in the process of control program design, and duty cycle can be calculated by the following formula,

$$\gamma = \frac{CCRx}{(ARR + 1)}(x = 1, 2) \tag{36}$$

where *CCRx* is comparison value, *ARR* is value of automatic reload register in the stm32 chip.

So the modified duty cycle $\gamma_2$ of the linear actuator $L_2$ can be obtained as:

$$\gamma_2 = \frac{V_2}{K_\gamma} = \frac{CCR1}{(ARR + 1)} + \frac{K_d(U_1 - U_2)}{K_\gamma t}. \tag{37}$$

According to the above analysis, the coordination control of the lifting mechanism can be completed.

## 4. Experimental Research of the Wheelchair-Stretch Assistive Robot

### 4.1. Experimental Research of the Walking Mechanism

In order to evaluate the performance of the compound position control method and verify the correctness of the kinematic equations, experimental research of the walking mechanism is conducted. First of all, the design parameters of the walking mechanism are shown in Table 2.

**Table 2.** Parameter value list of the walking mechanism.

| Parameter | Value | Parameter | Value |
|---|---|---|---|
| The velocity of the motor | 4900 r/min | Reduction ratio for reducer | 1:80 |
| Radius $R$ of the mecanum wheel | 63.5 mm | Bias angle $\alpha$ | 45° |
| $m_1$ | 272 mm | $m_2$ | 451.5 mm |

According these parameters, the corresponding jacobian matrix can be computed as:

$$J_1 = \begin{bmatrix} \frac{1}{63.5} & \frac{1}{63.5} & \frac{-723.5}{63.5} \\ \frac{-1}{63.5} & \frac{1}{63.5} & \frac{723.5}{63.5} \\ \frac{-1}{63.5} & \frac{1}{63.5} & \frac{-723.5}{63.5} \\ \frac{1}{63.5} & \frac{1}{63.5} & \frac{723.5}{63.5} \end{bmatrix}. \tag{38}$$

The motion profile exhibits 1.0 m parallel motions along the *y*-axis of the global coordinate frame. In addition, the trajectory consists of acceleration for 4 s followed by deceleration for 4 s. The profile is formed by a 5th order polynomial equation, as shown in Figure 22a. Based on the profile, Equations (5) and (37), the theoretical curves for the velocity of the motor could be obtained, as shown in Figure 22b.

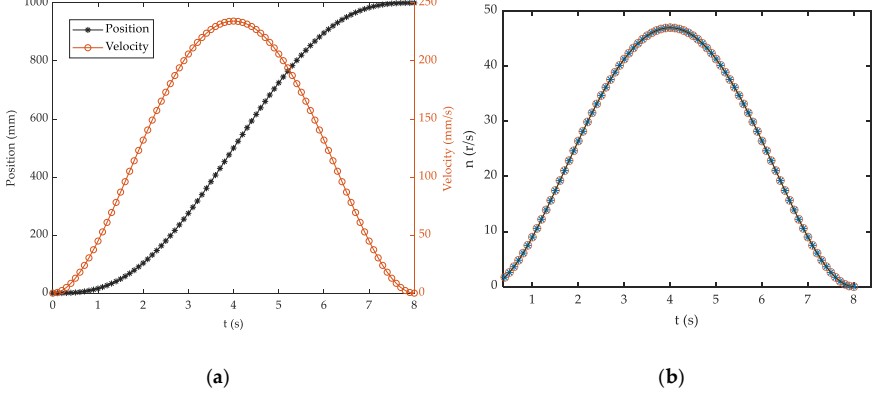

(a)　　　　　　　　　　　　　　　　　　　(b)

**Figure 22.** Trajectory planning of the walking platform and the motor; (**a**) trajectory planning for the center of the walking platform; (**b**) the motion curve for four motors of the mecanum wheel.

The following conclusions can be drawn from the Figure 22b:

(1)    In the mobile process, the trajectories of the four motors for the mecanusm wheel are the same, which is identical to the theoretical analysis.

(2)    The maximum velocity of the motor is less than the rated velocity (4900 r/min ≈ 81.6 r/s) and the motion curve is suitable to the experimental requirements.

The simulation data is processed, and the corresponding binary files can be obtained. Then, the data are imported into the control system to control the robot's motion in the experimental environment under a normal position control method, which means position control without the feedback function) and compound position control method, as shown in Figure 23a. Data collection is conducted every 100 ms, so 80 data samples can be obtained during the whole movement (100 ms × 80 = 8 s). These data can be processed to obtain the real position of the robot under normal conditions and under composite position control, as shown in Figure 23b.

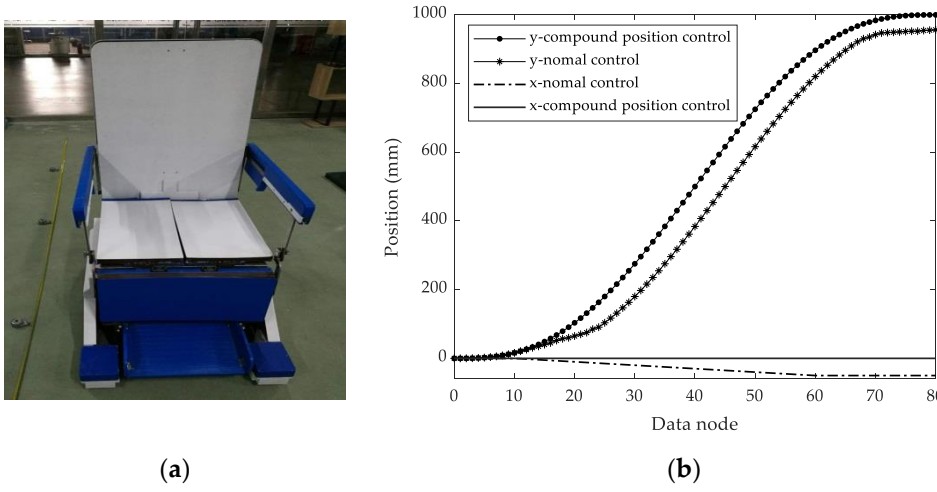

(**a**)                                    (**b**)

**Figure 23.** The walking test experiment of the robot; (**a**) test environment of the robot; (**b**) the velocity change of the linear actuators $L_1$ and $L_2$.

Observing Figure 23b, the following conclusions could be obtained:

(1)    The waveform for the normal position control shows steady-state positioning error not only for the *x*-axis but also for the *y*-axis. On the contrary, the compound position control can achieve a successful result in no steady-state error.

(2)    In data node 80, the position of the robot for the compound position control is 995.6 mm and there is a 4.4 mm deviation from the theoretical value 1000 mm.

Based on the above experiment analysis, we can understand that the compound position method is not suitable for the precise control requirements, but it is enough for the control of the walking mechanism of the stretcher-wheel assistive robot as control accuracy is undemanding.

### 4.2. Experimental Research of the Folding Mechanism

The driving device of the folding mechanism is shown in Figure 24. In order to verify the reliability of the folding mechanism and the feasibility of the driving device, different weight experimenter is used for verification. First, the experimenters lay flat on the wheelchair-stretcher assistive robot, and then the folding mechanism is transformed, as shown in Figure 25. The whole moving process is stable and the mechanism folds smoothly, which indicates that the force of the torsion spring plays a role in balancing the weight of the equipment and the human body. Since the change of current reflects the change of force, the current equipment is used to test the change of motor current. The test curve is shown in Figure 26, and the following three conclusions can be drawn.

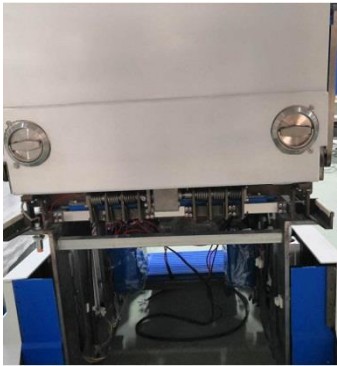

**Figure 24.** The driving device of folding mechanism.

(1)  From the perspective of the curve, it is basically consistent with the variation law of the motor torque curve simulated in Figure 11, which verifies the correctness for the theoretical analysis of the driving device.

(2)  The rated current of the motor is 8.3 A. When the tester is 90 kg, the actual maximum current measured is 8.26 A, so the maximum carrying capacity of the whole equipment is 90 kg.

(3)  At the later stage of the movement process, the motion curves fluctuate obviously, because the experimental personnel adjusted their posture, resulting in fluctuations in the force applied on the back. when the back of the wheelchair is folded from a horizontal position to a nearly vertical position, the center of gravity slips relatively during the folding process because the rotation center of the hip joint and the back of the wheelchair do not coincide, which causes the body to lift frequently.

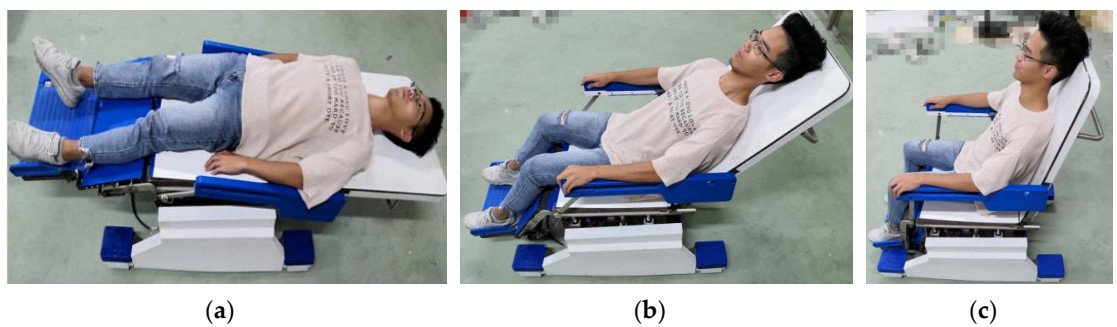

|      (**a**)      |      (**b**)      |      (**c**)      |

**Figure 25.** Folding process of the wheelchair-stretcher robot; (**a**) lying on the back posture; (**b**) in the folding process; (**c**) end of folding.

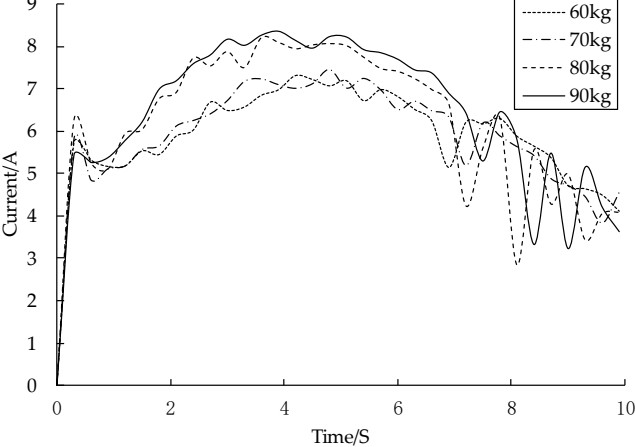

**Figure 26.** Current of the motor at different weights.

*4.3. Experiment Research of the Lifting Mechanism and the Toilet Mechanism*

(1) Experiment Research of the Lifting Mechanism

The automatic lifting function experiment mainly tests the synchronization of the robot in the process of rising and fall on both sides. The experimental height is set in the range 0–100 mm; the curve is formed by the fifth order polynomial equation and the motion time is 8 s. The position and velocity curves of the lifting mechanism under the theoretical state can be obtained, as shown in Figure 27. The performance parameters of the milang electronic ruler and linear actuator are shown in Table 3. By comparing the parameters in Figure 27 and Table 3, it is found that the planned trajectory is within the range of motion.

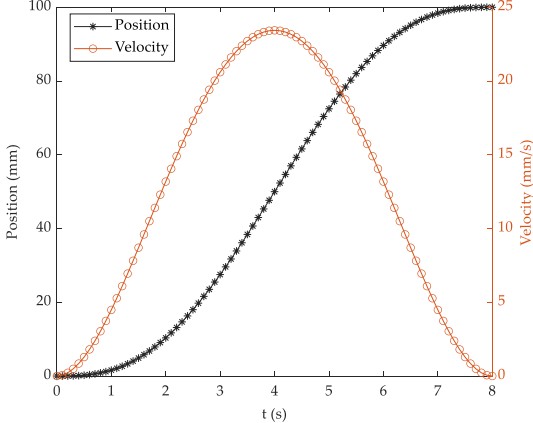

**Figure 27.** Trajectory planning of the lifting mechanism.

**Table 3.** Performance parameters of the milang electronic ruler and linear actuator.

| Name | Voltage | Range | Maximum Velocity | Resolution Ratio | Output Type |
|------|---------|-------|------------------|------------------|-------------|
| Milang electronic ruler | 12~24 V | 0~300 mm | 10 m/s | 0.01 mm | 0~5 V |
| Linear actuator | 24 V | 0~300 mm | 25 mm/s | — | — |

The position close loop control method is used to control the motion of the lifting mechanism, and the output voltage of the milang electronic ruler could be measured by applying the A/D conversion function in the STM32 chip. The sampling time is 10 ms, and the average voltage is calculated by collecting 10 times of data. The position difference between linear actuator $L_1$ and $L_2$ can be obtained by the conversion of the average voltage, and the value of linear actuator $L_2$ is modified.

The time interval of each updating data value is 130 ms; therefore, in the process of the whole movement, a maximum of 61 data samples could be saved. The curves of the position and velocity are formed by using 61 data samples, as shown in Figure 28.

Observing Figure 28, the following conclusions could be obtained:

(1)   The real position trajectories of the linear actuators $L_1$ and $L_2$ are stable and fluctuation is not obvious. The position close loop control strategy is suitable for the lifting mechanism.
(2)   To ensure that the position of the linear actuator $L_1$, the velocity of the linear actuator $L_2$ slightly larger than the velocity of the linear actuator $L_1$.

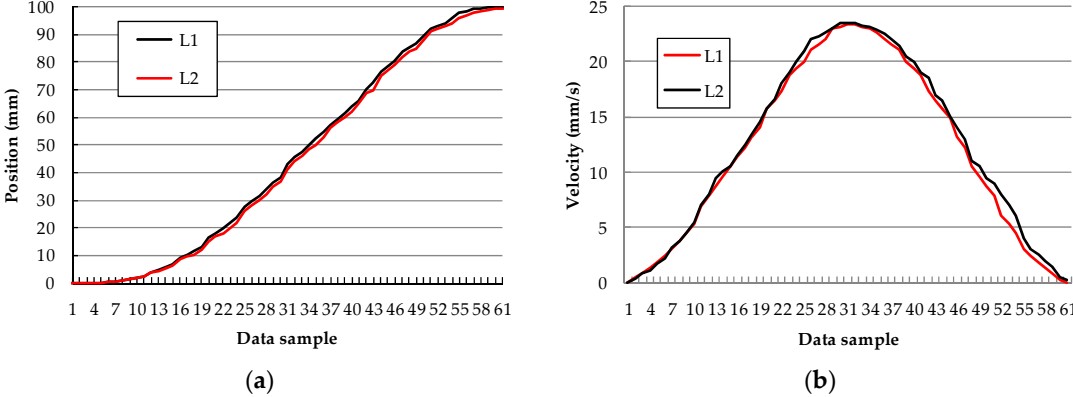

(a)　　　　　　　　　　　　　　(b)

**Figure 28.** The position and velocity change in test experiment of the lifting mechanism; (**a**) the position change of the linear actuators $L_1$ and $L_2$; (**b**) the velocity change of the linear actuators $L_1$ and $L_2$.

In addition, tests also are conducted by multiple groups of passengers and the results show that the robot runs smoothly and passengers do not feel the deviation of movement as shown in Figure 29. It also indicates that the control algorithm of the lifting mechanism is in line with the design requirements.

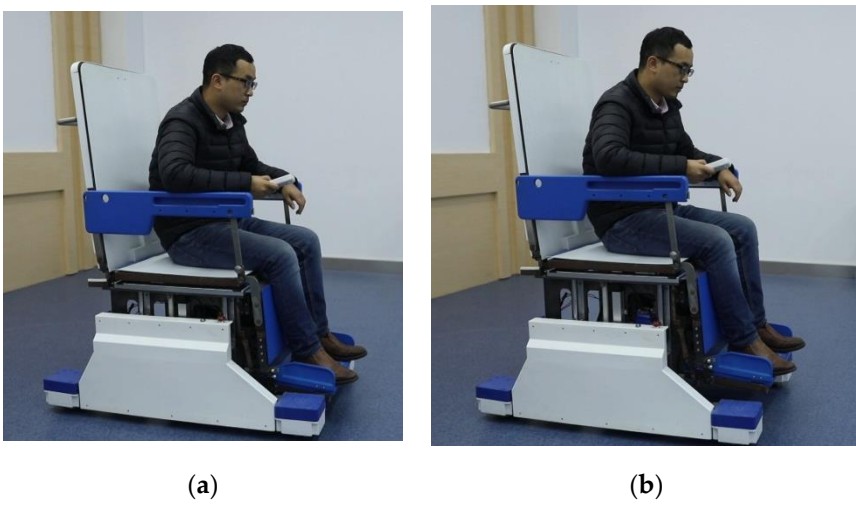

(a)　　　　　　　　　　　　　　(b)

**Figure 29.** Experiment research of the lifting mechanism; (**a**) state 1 in the lifting process; (**b**) state 2 in the lifting process.

(2) Experiment research of the toilet mechanism

The control of the toilet mechanism is relatively simple. In order to test its function, the current test method is used to detect its movement process. Firstly, passengers on the robot stand up by relying on the armrest, and then hold the open button on the remote control for the opening and closing of the toilet mechanism, as shown in Figure 30a,b.

In the process of moving, the current values of the left and right linear actuators of the toilet mechanism are recorded, and the corresponding experimental results as shown in Figure 30c. Observing Figure 30c, the following conclusions could be obtained:

(1)　Excepting for the instantaneous impulse current in the initial stage, the curve of the left and right linear actuators is smooth in the process of moving.

(2)　In the opening process of the toilet mechanism, the current increases gradually; in the closing process of the toilet mechanism, the current decreases gradually. The reason for this phenomenon is mainly the lengthening of the force arm of the linear actuator.

(3)  Due to machining and assembly errors, the friction force of the slide rail is increased; therefore the current curves of the left and right linear actuators do not coincide with each other in the process of moving.

So we know that the motion of the two pads is smooth and doesn't interfere with the handrail bracket, which verifies the design function can meet the requirements.

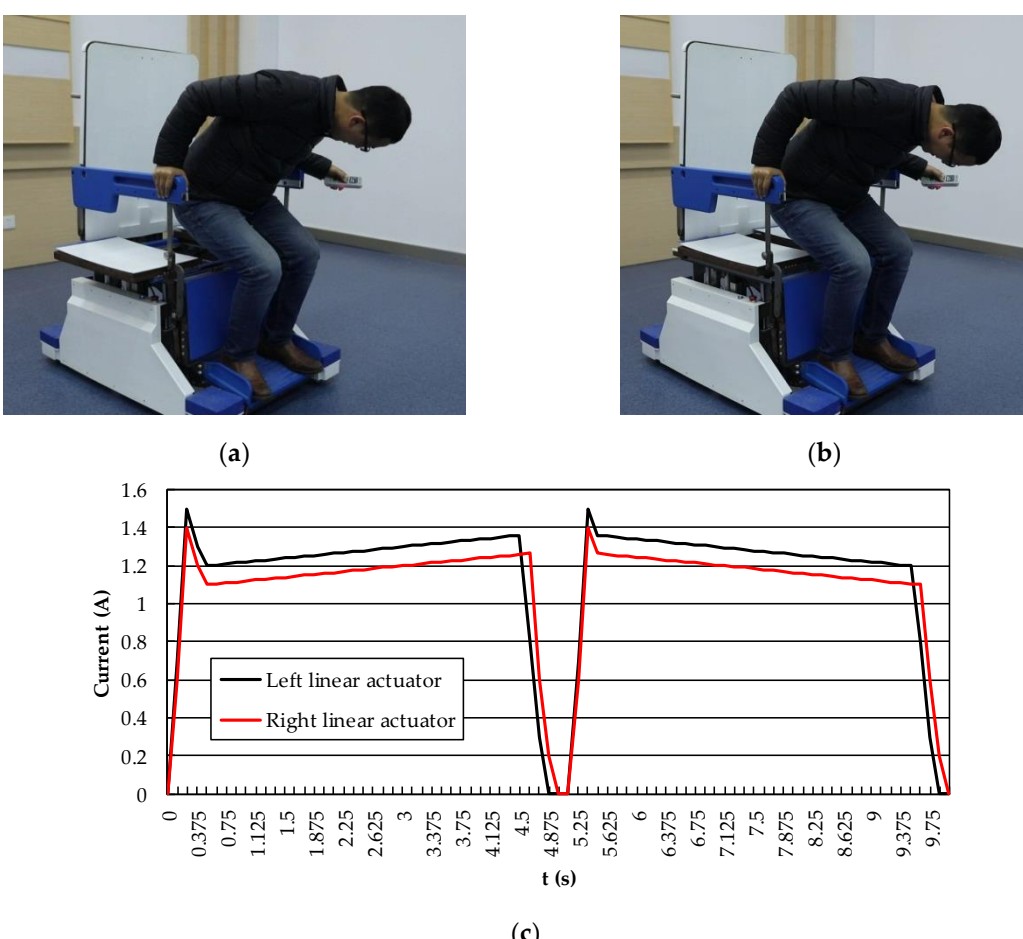

**Figure 30.** Experimental research of the toilet mechanism; (**a**) opening process of the toilet mechanism; (**b**) closing process of the toilet mechanism; (**c**) the current change curves of the left and right linear actuators.

### 4.4. Experimental Research of Cooperating with Other Equipment

The robot is used together with the transfer equipment to test its docking ability, as shown in Figure 31.

First of all, the wheelchair-stretcher assistive robot is controlled by the remote control to move to the bedside, and then the height of the lifting mechanism is adjusted to achieve high docking with the bed, as shown in Figure 31a. The transfer equipment is used to move the object from the bed to the wheelchair-stretcher assistive robot; when the object fully reaches the wheelchair-stretcher assistive robot, the transfer equipment is moved back to the bed, as shown in Figure 31b. The folding mechanism is used to transform the object from the lying position to the sitting position, as shown in Figure 31c.

From the operation process in the figure, it can be seen that the transfer equipment can accurately move the object to the wheelchair-stretcher assistive robot without interference in the movement process, the docking is accurate, and the folding mechanism can also realize the automatic folding function.

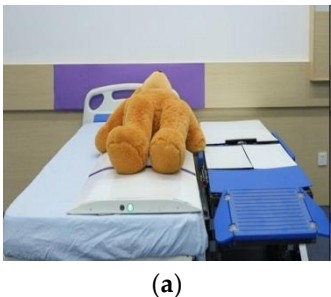 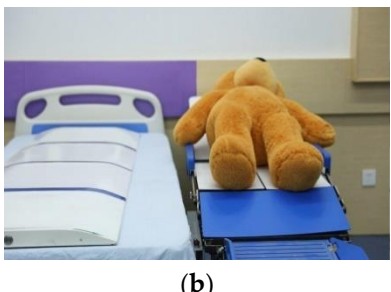 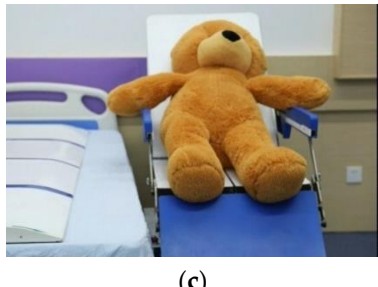

(**a**) (**b**) (**c**)

**Figure 31.** The movement process of the object from the bed to the wheelchair-stretcher assistive robot; (**a**) the transfer equipment carries the object to the wheel-stretcher assistive robot; (**b**) the object is moved onto the wheel-stretcher assistive robot and the transfer equipment return; (**c**) the robot changed from a stretcher to a wheelchair.

## 5. Conclusions

A novel wheelchair-stretcher assistive robot is analyzed, computed and designed in detail in this paper; finally, the following conclusions are obtained:

(1) The kinematics equation and control equation of the walking mechanism are established, and the correctness of which is determined by theoretical calculation and practical experiment.

(2) A five-link mechanism with single degree of freedom is proposed to realize the folding motion of the robot. The optimal design of the parameters of the mechanism is realized by using the minimum conclusive area method. By studying the force properties of the folding mechanism, the torsion spring and RV reduction motor are used as the driving device, the output torque of the motor is reduced, and the motion optimization and mechanical optimization of the folding mechanism are completed.

(3) Combined with ergonomics, the mechanical structure design of four functional modules is completed. The movement of each module is verified by the experiment research.

(4) Based on the STM32 chip, the control system of the whole prototype is achieved, and the synchronous control algorithm for the control of the lifting mechanism and the compound position control algorithm for the walking mechanism are applied. According to the experimental results, these algorithms can meet the motion requirements. Based on the current, the load capacity of the robot is determined as 90 kg.

In future research, we will reduce the weight of the equipment and improve its reliability. In addition, in the whole folding process, the joint of the human body is not concentric with that of the wheelchair stretcher assistant robot, which results in poor comfort of the patients, so these problems will be further studied in the follow-up work.

**Author Contributions:** Conceptualization, L.S., M.Y., Z.G., and H.W.; methodology, L.S., J.F., and F.D.; software, L.S. and Y.T.; validation, L.S. and Y.T.; formal analysis, M.Y.; investigation, L.S.; resources, L.S. and M.Y.; Writing—Original draft preparation, L.S. and Y.T.; Writing—Review and editing, M.Y., Z.G., F.D., J.F., and H.W.; project administration, L.S.; funding acquisition, L.S. All authors have read and agreed to the published version of the manuscript.

**Funding:** This research was funded by General Scientific Research Project of Education Department of Zhejiang (grant number Y201840178), Ningbo natural science foundation (grant number 2019A610123) and postdoctoral program of Zhejiang province (grant number ZJ2017032).

**Conflicts of Interest:** The authors declare no conflict of interest.

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
