# Peer review of "Analysis, Design, and Experimental Research of a Novel Wheelchair-Stretcher Assistive Robot"

_applsci, doi:10.3390/app10010264_

Round 1

Reviewer 1 Report

The results reported, underline the potential of the use of robotic devices for medical and care applications. The paper is well-written, concise, and easy to understand. It would be interesting to have some information on the estimated cost of production and the market price of such a device that could change the lives of many.

Author Response

Thank you for your suggestion and confirmation. We will continue to strive for further improvement in product intelligence.

Reviewer 2 Report

The topic covered in the paper is actual and this actuality is well presented in the introduction of the paper. What I miss in the introduction of the paper are more references dealing with control of complex electro-mechanical devices such as robotic arms. I also miss some methodological contribution of the design, aims of which should be in my opinion defined in the introduction of the paper, referencing some other works in the area published in renown journals.

This relates to my second remark and that is the intelligence of the presented approach. The wheelchair presents certainly an impressive device, but what is intelligent on it should be defined in a clearer way. This should be defined in introduction of the paper and also concluded if the aims for giving some intelligence to a wheelchair have been achieved and what needs to be done to increase the level of its intelligence further.

In the second chapter it is not clear in my opinion how the models described in the chapter 2.2.1 and 2.2.2 are interconnected with the design phase shown in the chapter 2.3. I am also missing the Chapter 2.1.

I also have the feeling that the main part of the chapter 2 about the ergonomics of the design is also not connected well to the sub-chapters 2.2.X and 2.3. Intelligent elements of the design are not emphasized and not considered in the chapter.

I like the chapter about the control design and positively evaluate the applied speed/position control algorithm. However I miss some results of the achieved control quality in terms of simulation experiments and practical tests. If it is possible, please add at least some quantitative results of the designed control system in form of graphs and data evaluating the control quality and effects of the performed optimizations. Again intelligence of the system should be defined/emphasized (is the control approach somehow adaptive/robust)?. 

These results of the control tracking could be also presented in the Chapter 4. Some quantitative analysis of the obtained control quality and quality of the actuation should be given in the chapter. There is one graph which is showing the course of the current, some oscillations for heavier weight seem to be there, which could translate to some oscillations in movement of the wheel-chair or not, this should be presented in the article in my opinion. I find these results as scientifically lacking.

The same goes for other mechanisms, it is very nice to have photographs of the objects being used with the wheelchair, however some graphs illustrating the positioning of the chair and its elements should be presented. 

In general I am not sure about the methodological scientific contribution of the paper. I think this aspect should be improved throughout the paper as well as the intelligent element and how did the ergonomic aspects of the design directly influence the resulting construction of the wheelchair. 

Author Response

Dear reviewer:

        Thank you very much for taking time out of your busy schedule to read my paper.  And Thank you for your suggestions. According to your suggestions , we have revised the paper.

        The attached file is the author's specific response to each comment.

Reviewer 3 Report

The paper has merits but it needs to be improved from contents, editing and language point of view. In the following, it can be find some of the identified issues:
- Line 16: it would be recommended to express those functionalities as one expression (eg. something like current physiological needs assistance)
- one or two more keywords can be included;
- please check the misspelling and typing errors (see. line 32: proper case for "People" word, line 111 "Mecanum", line 137 "components ad seven", in line 162 check the symbol following alpha/same in line 319 after L1, line 321 after U1, line 168 "Combing", line 173 "iwhen", line 209 "compution", in fig. 17 "Linear acutator");
- line 36: "elderly are eager to go to the toilet" could be something related to "needs";
- In Introduction there are presented some developed solutions related to the transport devices, but authors' comments on their shortcomings reported to the current contribution are missing;
- In section 2, some references to standards related to the mentioned types of robots design and implementation can be inserted (see line 76 "the relevant standards of medical devices");
- in eq. 5, please check if it is oportune to specify the measurement unit;
- line 132 - please check and rephrase: "So the mechanism in shown in figure 7 ...";
- line 138 - please check the term "planer link";
- please insert in the text the term "F" after its corresponding description (eg. "degree of freedom (F)"). Also, as with the F is denoted the force, an alternative notation is recommended;
- line 204-205: please check and rephrase "Using MATLAB software and combining equations (15)-(21), ..." as the current statement is confusing (in a concise form the Matlab usage is recommended to be clarified);
- line 206-207: please consider to revise some of the used terms (eg. "large ... torque/power", "high" term is suggested to be considered);
- Fig. 11 seems having too low resolution;
- please check and revise the term "walking" ("walking mechanism" to a device on wheels seems to be forced/inappropriate);
- line 311: please check the term "position of the theory" (the term can be refered as "reference value of ..." and then specifing how it is computed);
- line 310: please denote the figure;
- line 315: please check the statement "... the modified PWM value can be obtained through the transform ..." (transformation/conversion?)
- line 316: the statement "The process is as follows:" is unfinished;
- line 319: please check the statement: "Assuming the numbers of the linear actuator is L1, L2, and L1 is benchmark reference.";
- line 323: check and update: "... the proportional coefficient is assumed to Kd ...";
- line 327: unfinished statement, please consider to revise;
- lines 335/336/372/375/389/391: there is about "experimental research"?
- lines 344-346: there should be explained with which degree the experimental results are consistent with the theoretical results;
- lines 373-374: statement needs to be revised;
- Please consider to insert a statement in the Conclusion section before he existing list;
- As some controllers are designed and implemented, their performances (using specific indices) are recommended to be specified.

Author Response

Thank you for your suggestion. According to these suggestions, we have revised the paper. and the attached file is the corresponding answer for point-to-point.

Round 2

Reviewer 2 Report

The paper has been considerably improved and all my remarks have been sufficiently addressed in the paper. Major parts of the paper have been re-written and are considerable improvement over the previous version. I therefore suggest the paper to be accepted in the present form after some minor corrections, which I am suggesting further:

As a minor remark and I am leaving it up to consideration of the authors,  I would suggest only a bit more detailed evaluation of the ergonomics principles applied in the design of the wheel chair in order to extend the chapter 2 a bit. Please also check annotation of the X axis in Figure 28, I would suggest use the word sample instead of "Number of data" or change the x-axis so it would show time. In the text it could also be defined as samples instead of sets of data (this suggests it is a multi-parametric data vector). Figure 26 caption should probably read Current of the motor at different weights.  Please re-consider putting the graphs without green background into the paper (Figs 26,28,30). I am not a native English speaker, but I feel that the language in the paper is also in a need of grammar polishing. 

Author Response

Dear reviewer:

      Thank you very much for taking the time to review our paper. According to your suggestions, we improve our paper. the attached file is the authors' response by point to point.
